# Prevalence, Incidence, and Surgical Treatment Trends of Cerebral Palsy across Türkiye: A Nationwide Cohort Study

**DOI:** 10.3390/children10071182

**Published:** 2023-07-07

**Authors:** İzzet Özay Subaşi, İzzet Bingöl, Niyazi Erdem Yaşar, Ebru Dumlupinar, Naim Ata, M. Mahir Ülgü, Şuayip Birinci, Mustafa Okan Ayvali, Serkan Erkuş, Mehmet Salih Söylemez, Güzelali Özdemir

**Affiliations:** 1Department of Orthopedics and Traumatology, Faculty of Medicine, Erzincan University, Erzincan 24002, Türkiye; 2Department of Orthopedics and Traumatology, Oncology Training and Research Hospital, Ankara 06230, Türkiye; dr.izzetbingol@hotmail.com; 3Department of Orthopedics and Traumatology, Ankara Bilkent City Hospital, Ankara 06800, Türkiye; erdem__yasar@hotmail.com (N.E.Y.); drguzelali@yahoo.com (G.Ö.); 4Department of Biostatistics, Faculty of Medicine, Ankara University, Ankara 06230, Türkiye; eedumlupinar@gmail.com; 5Ministry of Health, Ankara 06430, Türkiye; naim.ata@saglik.gov.tr (N.A.); mahir.ulgu@saglik.gov.tr (M.M.Ü.); suayipbirinci@gmail.com (Ş.B.); mustafaokan.ayvali@saglik.gov.tr (M.O.A.); 6Department of Orthopedics and Traumatology, Medifema Hospital, İzmir 35860, Türkiye; s.erkus@ymail.com; 7Department of Orthopedics and Traumatology, Umraniye Training and Research Hospital, Istanbul 34764, Türkiye; slhsylmz@gmail.com

**Keywords:** cerebral palsy, paediatric, incidence, surgical management

## Abstract

Background: Cerebral Palsy (CP) is the most prevalent neurodevelopmental disorder in childhood. Our aim is to identify the demographics of CP in Turkish children in addition to clinical associations and surgical preferences. Methods: Based on national health system data and the International Classification of Diseases (ICD)-10 code for CP, data were evaluated from a total of 53,027 children with CP born between 2016 and 2022, and 9658 of them underwent orthopedic surgery in those years. The incidence and frequency of CP were assessed for the parameters of age and gender. Age at the time of surgery; codes pertaining to surgical interventions; and regions, cities, and hospitals where diagnoses and surgical procedures were performed were also evaluated. Results: There were 29,606 male (55.8%) and 23,421 (44.2%) female patients. The diagnoses of the patients were mostly (76.1%) performed in secondary and tertiary hospitals. The prevalence of CP among children in 2016–2022 was estimated to be 7.74/1000 children. The minimum and maximum incidence rates of cerebral palsy among children between 2016 and 2022 were calculated to be 0.45 and 1.05 per 1000, respectively. Tenoplasty–myoplasty tendon transfer operations were the most common surgeries (47.1%). Conclusion: CP remains a significant health challenge, underpinning a considerable proportion of childhood motor dysfunction. A dedicated national registry system for CP focused on classifying the condition, streamlining treatment, and tracking outcomes would be a valuable tool in our collective efforts to address this critical issue more effectively.

## 1. Introduction

Cerebral palsy (CP) is an umbrella term for a group of motor dysfunction disorders caused by permanent and non-progressive lesions of the developing brain [1]. CP is the most prevalent neurodevelopmental disorder in childhood, with an estimated prevalence of 2–3 per 1000 live births worldwide [2,3,4]. Secondary impairments such as joint deformities, muscle contractures, hip dislocations, and scoliosis further complicate the motor dysfunction of children with CP. Secondary impairments significantly limit the participation of children with CP in daily activities [5,6,7]. Due to multiple difficulties, children with CP frequently require specialized health, social, and educational services.

A combination of delayed motor milestones, abnormal neurologic examination, the persistence of primitive reflexes, and abnormal postural reactions are indicative of a potential diagnosis of CP. In addition, the neonatal medical history can potentially elicit concerns and requests for heightened monitoring [8,9,10]. Sequential assessments are crucial in tracking advancements in cases where cerebral palsy is suspected. The identification of brain damage is typically simplified when it is supported by documented evidence from cranial ultrasound, computed tomography (CT scan), or magnetic resonance imaging. The amalgamation of natal history, risk factors, and physical examination findings determines Türkiye’s diagnostic requirements for CP. In our country, the initial stage of diagnosing cerebral palsy (CP) involves the contribution of neonatal physicians who assess the risk factors during the perinatal period. Following this timeframe, family physicians and paediatricians facilitate the referral of patients to paediatric neurologists for the purpose of obtaining additional diagnostic assessments to confirm the presence of the disease. Following the establishment of a conclusive diagnosis, the patient’s treatment plan typically involves medical professionals specializing in physical therapy, orthopedics, and neurosurgery. Typically, children with more severe symptoms are often diagnosed by the age of six months, while milder cases may require a longer period for diagnosis.

Comprehending the attributes and prevalence of cerebral palsy (CP) holds significant importance in effectively managing the potential comorbidities that may arise as a result of this condition. Multiple factors, including the availability of medical services, the level of medical knowledge and technology, and the socio-economic status of each country, influence the characteristics and prevalence of CP [1,11,12]. While the majority of data on the characteristics and prevalence of CP come from Europe, Australia, and North America, few prevalence studies have been conducted in Türkiye, and none are representative of the national prevalence rate [13].

Various forms of orthopedic surgery can be employed to treat the muscular and skeletal deformities that arise as a secondary effect of brain injury in CP [14,15,16]. Although the studies evaluating the nationwide cohort related to cerebral palsy find themselves in the literature, there was no study carried out in the nationwide cohort investigating the surgical treatment trends of cerebral palsy [6,17]. Furthermore, the available reports fail to address all age groups, types of surgical interventions, and distributions of cases by sex and institution type [15,18,19].

As the course of CP becomes more unique for each patient, treatment options become accordingly diverse, and it becomes more difficult to establish standard treatment protocols. Reviewing national data may give physicians a clearer understanding of CP diagnoses and treatments together with improved guidance. In addition, evaluating, categorizing, and determining the frequency of surgical procedures performed for these patients will assist orthopedic surgeons in identifying surgical trends. This study aims to identify the demographics and surgical preferences of CP cases among Turkish children.

This retrospective cohort study was conducted in accordance with the Declaration of Helsinki and received approval from the Turkish Ministry of Health with a waiver of informed consent for retrospective data analysis and the health information privacy law (ID: 95741342-020/27112019). The Turkish Ministry of Health’s database was used to compile the health records of individuals younger than 18 years old who were admitted to public, private, and university health institutions. In Türkiye, every citizen has a unique civil registration number, which is used in all healthcare interactions and facilitates seamless connections between healthcare registries. All patient information is stored in an electronic database known as e-Nabız [20].

### 1.1. Registers and Study Cohort

Based on e-Nabız data and the International Classification of Diseases (ICD)-10 codes for cerebral palsy (G80) and its seven subdivisions (G80.0: spastic quadriplegic cerebral palsy; G80.1: spastic diplegic cerebral palsy; G80.2: spastic hemiplegic cerebral palsy; G80.3: dyskinetic cerebral palsy; G80.4: ataxic cerebral palsy; G80.8: cerebral palsy. Others: G80.9 cerebral palsy, undefined), a search was conducted for records between January 2016 and December 2022. To determine the population of patients who had undergone surgical interventions, the Turkish Ministry of Health codes known to be applied for patients with CP were selected (https://skrs.saglik.gov.tr/, accessed on 11 March 2023). This was carried out to facilitate the processing of data. All outcome variables were extracted and collected from the medical records of patients as available in the e-Nabız database. Patients who did not have sufficient data, such as limited data transfer to e-Nabız from the health institution they applied to, and those who did not allow open access to e-Nabız data were excluded from the study.

As of 2022, the total number of children under the age of 18 who have been diagnosed with cerebral palsy in our nation amounts to 184,311. The prevalence calculation was conducted utilizing the provided dataset. In order to calculate the incidence of cerebral palsy, the researchers utilized the ratio between the number of newly diagnosed patients with cerebral palsy registered with e-Nabız from 2016 to 2022 and the corresponding population under the age of 18 for each respective year. This study aims to evaluate various demographic characteristics, including the geographical region where the diagnosis was made, by using the dataset consisting of 53,027 people born in our country and diagnosed with cerebral palsy between 2016 and 2022. The main reason for this situation is that regular data transfer to e-Nabız, the patient registration system of our country, started in 2016. Furthermore, an analysis was conducted on the data of 9658 children who underwent surgical intervention between 2016 and 2022. It should be noted that the data from the registry system were easily accessible during this period.

### 1.2. Patient Groups

Age, gender, and ethnic origin were noted as patient-related characteristics. Public hospitals, private hospitals, and public and private university hospitals were the considered types of institutions. The main outcome measures assessed the incidence and frequency for the parameters of age and the gender distribution of CP. Türkiye comprises seven distinct geographical regions, each of which exhibits different socio-economic characteristics, namely Marmara, Central Anatolia, Aegean, Mediterranean, Southeast Anatolia, Eastern Anatolia, and Northern Anatolia. These regions exhibit variations in several dimensions, including socio-economic advancement and availability of healthcare facilities. The urban areas of the nation, which serve as significant hubs for industry and commerce, are situated in the Marmara, Aegean, Central Anatolia, and Mediterranean regions. In contrast, the rural regions of Eastern Anatolia, Southeast Anatolia, and Northern Anatolia are predominantly agricultural. Due to this rationale, while primary healthcare services are available in Turkey’s Eastern, Northern, and Southeast regions, facilities in other areas are favoured for complex diagnoses and treatments of various health conditions, such as cerebral palsy. Consequently, the geographical areas in which the patients within our study cohort received both their diagnoses and surgical interventions were identified and analysed.

The patients treated surgically were divided into four sub-groups according to the ages of 0–3 years, 4–6 years, 7–10 years, and >10 years. The primary rationale behind categorizing the patients based on their age pertains to the unavailability of the GMFCS scores of the patients. The grouping of patients based on age can be informative in assessing surgical interventions given that those with a higher GMFCS score tend to require such procedures at an earlier stage. During the establishment of these groups, we derived valuable insights from the research conducted by Telleus et al. in their 2022 publication [15]. Nevertheless, due to the utilization of Swedish registry data and the incorporation of patients’ GMFCS scores, the direct applicability of this study was limited. When the conclusions derived from this study are integrated with our routine paediatric and orthopaedic experiences, it becomes evident that a surge in surgical interventions is observed after the initial three years of age, irrespective of GMFCS scores. Furthermore, there is a notable increase in soft tissue operations and BTX injections between the ages of 4 and 6. Additionally, surgical interventions aimed at averting permanent deformities are more prevalent during the ages of 7–10. Given that our focus primarily revolves around deformity correction and joint-sparing surgeries for individuals aged ten years and above, we have proceeded with the categorization process. The codes pertaining to surgical interventions were categorized into four distinct sub-groups of botulinum toxin injection, joint reconstruction (arthrodesis, surgical reduction in joint dislocations, surgical releases of joint contractions, etc.), tenoplasty–myoplasty tendon transfer, and deformity correction (distal femoral extension osteotomies, proximal femur derotation osteotomies, medial column corrections of the foot, etc.), and these were evaluated in terms of the regions and cities where the surgical procedures were performed and the gender and age of the patients. The study assessed the most invasive surgical procedures conducted on paediatric patients who underwent multiple surgical interventions. To clarify, the documentation of patients who received repetitive botulinum toxin injections from 2016 to 2022 was categorized as botulinum toxin injection interventions. However, if a patient underwent a procedure such as a deformity correction or osteotomy during the same time frame, the invasive procedure was incorporated into the surgical assessment.

### 1.3. Statistical Analysis

Descriptive statistics were presented as mean ± standard deviation or median (minimum-maximum) according to the assumption of normal distribution for quantitative variables. Qualitative variables were presented as frequency (percentage). The assumption of the normal distribution of quantitative data was analysed with the Shapiro–Wilk test. The qualitative data of patients were evaluated using the Pearson chi-square test and Fisher’s exact test. Statistical significance was accepted at *p* < 0.05, and IBM SPSS Statistics 25.0 (IBM Corp., Armonk, NY, USA) was used for analysis.

## 2. Results

The present study analysed the demographic information of a sample of 53,027 children born within the years of 2016–2022. There were 29,606 male (55.8%) and 23,421 (44.2%) female children within this sample. While 91.2% of these patients were Turkish citizens, 8.8% were immigrants. It was observed that the majority of the diagnoses of the patients (76.1%) were carried out in secondary or tertiary hospitals affiliated with the Turkish Ministry of Health or in university hospitals. The demographic characteristics of the patients are given in Table 1.

While the number of cerebral palsy patients under the age of 18 in Türkiye in 2022 was reported as 184,311, the total population of individuals under the age of 18 in the country is estimated to be 23,823,305. According to the available data in 2022, the prevalence of cerebral palsy in individuals under the age of 18 was determined as 7.74 per 1000. While calculating our incidence rates, we carried out the calculation by dividing the number of children registered to the e-Nabız system by the diagnosis of cerebral palsy in the year in which the incidence was calculated relative to the population under the age of 18 in that year. In the annual incidence calculations between 2016 and 2022, the annual incidence was 1.05/1000 in 2016, 1.03/1000 in 2017, 0.86/1000 in 2018, 0.73/1000 in 2019, 0.45/1000 in 2020, 0.49/1000 in 2021, and 0.47/1000 in 2022. Table 2 shows annual incidences, populations under 18 each year, and annual diagnoses of CP in children under 18 years of age across Türkiye from 2016 to 2022.

At least one surgical intervention between 2016 and 2022 was performed for 9658 of the patients in our study cohort. It was observed that 5673 of the patients (58.7%) who underwent surgical intervention were male. While university hospitals ranked first among the health institutions where surgical interventions were performed for these patients (40.5%), interventions in secondary and tertiary hospitals affiliated with the Ministry of Health and private health institutions were also common (32.9% and 22.1%, respectively). The demographic data of patients who underwent surgical treatment are provided in Table 3.

Among the geographical regions where the patients were diagnosed, the Southeastern Anatolia region ranked first, followed by the Marmara region (26.4% and 26.1%, respectively). The Marmara region ranked first among regions where surgical interventions were performed, followed by the Central Anatolia region (31.0% and 20.4%, respectively).

Tenoplasty–myoplasty tendon transfer operations ranked first among the types of surgical procedures applied (47.1%). This was followed by botulinum toxin injections (45.5%), joint reconstructions (4.7%), and deformity corrections (2.7%). The surgical procedures performed for patients are detailed in Table 4 according to age groups.

## 3. Discussion

In this study, we aimed to evaluate the prevalence and incidence of CP in a nationwide cohort. Additionally, we evaluated surgical treatment trends among different levels of hospitals and geographical regions. The study revealed notable variations in the geographic distribution of medical facilities where the diagnoses of CP in children and surgical interventions were carried out. The primary factor contributing to this situation is the uneven distribution of medical resources across urban and rural areas in our country. While there are no restrictions on diagnosing cerebral palsy in rural regions, surgical interventions and subsequent diagnostic procedures are predominantly accessible in urban centres with more advanced medical infrastructure.

CP is a disease that should be treated by many healthcare professional teams, including orthopaedic surgeons, as it requires a multidisciplinary approach, and it impairs the quality of life due to symptoms such as joint deformities and muscle contractures. CP is the most common cause of motor disability in childhood, with an incidence of approximately 1.5–3.0/1000 live births [3,6,13]. Similarly to most neurological disorders, CP occurs appreciably more frequently among male patients than females [3]. In our study, we found that the prevalence of CP was 7.74/1000, and it was more common among male patients. Despite a general downward trend in the birth rate within our nation, 1,035,795 live births were recorded in 2022. This figure encompasses the children of immigrants born within our country’s borders. The high prevalence of cerebral palsy in regions with high birth rates, specifically eastern and southeastern Anatolia, can be attributed to various factors, including limited access to health services, home births, challenges in accessing paediatric intensive care units, and consanguineous marriages. The incidence of a disease is a metric that quantifies the frequency of its manifestation, while prevalence is a metric that quantifies its overall presence. The incidence metric identifies new cases, whereas the prevalence metric provides information on new and pre-existing cases. Since the inception of the e-Nabız patient registration system in 2016, incidence data have been consistently recorded with high accuracy. Notably, the incidence rate was observed to be high during 2016 and 2017. However, in 2020 and 2021, the incidence rate declined, which can be attributed to the limited number of hospital admissions resulting from the COVID-19 pandemic. This is thought to be the primary factor contributing to the reduction in occurrence.

In many epidemiological studies in the literature, it has been reported that the mean age of the patients followed and included in study groups for CP is approximately 11 years, and the age at first surgery is approximately 4 years [14,18,21]. In accordance with the literature, we determined the mean age of the patients who were treated with surgical interventions to be 4.97 ± 0.4 years at the time of surgery.

The utilization of population-based registries that involve complete case acquisition is a viable method for capturing the situation of a particular disorder as it eliminates the inherent biases associated with consecutive series and convenience samples [22]. The definition of cases should be unambiguous and uniform, and the process of identifying cases should be thorough and involve various sources to ensure maximal inclusion. The implementation of geographic categories is imperative for population-based registries to overcome institutional barriers, such as those found among different hospitals, rehabilitation centres, and universities. In our study, a patient registration program called e-Nabız, which is overseen by the Turkish Ministry of Health, was used to collect patient data. Thus, access to the medical records of all individuals who applied to any health institution was made possible.

Despite the presence of an electronic infrastructure that facilitates the management of a sizable cohort within our country, the implementation of a nationwide registration system specifically for CP data has yet to be realized. Numerous countries have implemented nationwide registration systems for child protection [1,2,7,11]. Electronic data systems encompass all pertinent patient information. The abundance of data pertaining to the sub-characteristics of diseases can lead to confusion and pose challenges in the statistical evaluation of the acquired data. The implementation of registry systems that are tailored to the specific needs of the medical field can facilitate the acquisition of comprehensive data pertaining to the aetiology, epidemiology, and treatment modalities of diseases such as CP.

A considerable number of paediatric patients diagnosed with CP receive orthopaedic surgical interventions aimed at correcting secondary musculoskeletal deformities and gait abnormalities, with the ultimate goal of enhancing or preserving their mobility [18]. According to McGinley et al., the correlations between orthopaedic deformities, function, and gait and health-related quality of life and general quality of life remain unclear and do not follow linear patterns. Furthermore, it was asserted that minimally invasive procedures can be highly advantageous in augmenting the functionality of patients [23]. Regrettably, our study was unable to assess the functional outcomes of individuals with cerebral palsy who received surgical intervention. According to the existing literature, prioritizing the injection of botulinum toxin after the age of two is recommended [10,24]. It is important to note that the administration of botulinum toxin will have a restricted duration of a few months, necessitating the requirement for subsequent injections or alternative therapeutic interventions. Our study findings indicate a higher frequency of less invasive procedures, such as botulinum toxin injections and tenotomy–myotomy, in comparison to other surgical interventions. The reason why minimally invasive methods and BTX injections are used at young ages is to prevent more invasive and difficult surgical procedures if possible and to delay them if it is not possible. However, in some patients, unfortunately, it is not even possible to delay major surgical interventions.

The three most commonly utilized procedures in the literature are reported to be hip adductor tenotomy, hamstring lengthening, and lower-leg soft-tissue surgery [15,18]. The study conducted by Rehbein et al. investigated variations in anatomical regions and gender among patients who were categorized by Gross Motor Functional Classification Scale (GMFCS) scores. However, the authors did not provide the age group data of the patients [18]. Telléus et al. conducted an assessment of surgical procedures based on anatomical localization, age, and gender and did not observe any statistically significant differences [15]. In our study, the most frequently performed surgical interventions regardless of anatomical localization were tenoplasty–myoplasty tendon transfer operations (47.1%). Subsequently, botulinum toxin injections were administered in 45.5% of cases, while joint reconstructions and deformity correction procedures were performed in 4.7% and 2.7% of cases, respectively.

Our review of the existing literature did not identify any studies that have assessed surgical interventions performed for children with CP based on their respective age groups. According to reports, there is a correlation between higher GMFCS scores in children and a decrease in the age at which they undergo their first surgical interventions [15,18,21]. Moreover, the literature suggests that surgical interventions administered to children vary across different geographical regions [21]. The findings of the present study indicate that injections of botulinum toxin were predominantly applied for individuals aged between four and six years (44.9%), while tenoplasty–myoplasty tendon transfer operations were more commonly performed for children over 10 years of age (35.4%). Furthermore, it was noted that surgeons in our country commonly perform operative procedures, including joint reconstructions and deformity corrections, for paediatric patients who have surpassed the age of 10.

To date, no studies in the literature have been identified that compare variables such as the healthcare facility where surgical interventions were conducted and the relevant geographical locations. The observed dissimilarities between healthcare facilities and geographical areas in which the patients in our study cohort received their diagnoses and where surgical procedures were carried out are noteworthy. The findings of our study indicate that the diagnosis of cerebral palsy is predominantly concentrated in the southeastern region. This region is situated in predominantly rural areas and is distinguished as the geographical region exhibiting the highest birth rate within our nation. However, considering the volume of surgical procedures conducted, it becomes evident that urban areas take precedence. These findings underscore the importance of conducting requisite research to facilitate surgical interventions in regions with higher rates of CP diagnosis. Furthermore, the disparity between the number of surgical procedures performed for immigrant children (1.5%) and the proportion of immigrants among patients with a diagnosis of CP (8.8%) is noteworthy. The influx of refugees to our nation experienced an escalation after 2012. From this year, the national health system has assigned a registration number to legally recognized asylum seekers, enabling them to access free healthcare services. Regrettably, the extent to which these children can benefit from health services remains uncertain due to various social challenges and impediments, including linguistic barriers. We believe that the aforementioned scenario does not impact the calculations of incidence and prevalence. This would be an appropriate topic for further professional investigation.

Although we had a substantial cohort, this study was limited by its retrospective nature, potentially leading to missing or incomplete data. This could have affected the accuracy of the identified prevalence and incidence rates. Despite providing valuable insights into the prevalence of CP and treatment trends, this study does not provide a detailed analysis of the efficacy and outcomes of surgical treatments. Even though GMFCS scores are documented within the health institution’s data of the patient’s follow-up in our country, it is regrettable that there is a lack of a distinct data tab within the e-Nabız system, which would enable us to record this information for the entire national cohort that we evaluate. Consequently, the GMFCS scores of the patients within our study cohort needed to be more attainable. Despite its significance in patient follow-up, the e-Nabız system is currently in its nascent stage with regard to the follow-up of individuals with cerebral palsy. Regrettably, the current documentation remains inadequate in capturing crucial data points such as the GMFCS scores of patients, the anatomical regions targeted by surgical interventions, the various subgroups of surgical interventions, and any complications that may arise from the use of the e-Nabız system. Our study lacked data on the socio-economic status of the patients, which could impact the accessibility and quality of the received treatment. This is an important factor that future research should consider. We could not assess the role of multidisciplinary interventions for CP due to the lack of available data. Such data could shed light on physiotherapy, occupational therapy, and speech therapy as crucial aspects of CP management. Finally, the lack of a standard definition and diagnostic criteria for CP across different hospitals may have led to inconsistencies in the reported data. The primary factor contributing to this scenario is physicians’ varying expertise levels across different healthcare facilities in diagnosing cerebral palsy. Less experienced physicians may struggle to identify subtle clinical presentations of the condition. The implementation of a nationwide tracking system that incorporates comprehensive demographic information, socio-economic indicators, GMFCS scores, the anatomical regions targeted by surgical interventions, the various subgroups of surgical interventions, and any complications would be highly advantageous for healthcare providers and individuals with cerebral palsy in devising appropriate treatment plans and follow-up protocols.

## 4. Conclusions

In our study, the prevalence of cerebral palsy in children under the age of 18 was found to be 7.74/1000. The minimum and maximum incidence rates of cerebral palsy among children between 2016 and 2022 were calculated to be 0.45 and 1.05 per 1000, respectively. The majority of the surgical procedures performed are tenoplasty–myoplasty tendon transfer operations (47.1%) and botulinum toxin injections (46.5%). Although the centres in which patients are diagnosed have shown a uniform distribution throughout our country, surgical procedures are often carried out in urban regions.

CP remains a critical health issue leading to a significant degree of motor dysfunction in children and posing challenges for healthcare systems. Precision in patient identification coupled with well-timed therapeutic interventions will offer potential benefits that extend beyond improved patient outcomes, including substantial reductions in healthcare costs. To this end, the impending establishment of a national registry system for CP in our country represents a milestone in tackling this disorder. This system is anticipated to enable a better classification and understanding of CP, facilitate more streamlined treatment strategies, and ultimately lead to more positive outcomes for patients. The creation of this registry, we hope, will provide a foundation upon which future research and initiatives for CP can be built, opening new avenues for improved patient care and clinical decision making.

## Figures and Tables

**Table 1 children-10-01182-t001:** The demographic characteristics of the patients born between 2016 and 2022.

	N	%	*p*
**Gender**			*p* ˂ 0.001 ^a^
Male	29,606	55.8%	
Female	23,421	44.2%	
**Geographic Region**			*p* ˂ 0.001 ^b^
Southeast Anatolia	14,010	26.4%	
Marmara Region	13,816	26.1%	
Mediterranean	7752	14.6%	
Central Anatolia	6456	12.2%	
East Anatolia	4062	7.7%	
Aegean Region	4027	7.6%	
North Anatolia	2903	5.5%	
**Healthcare Provider**			*p* ˂ 0.001 ^b^
2nd/3rd levels affiliated.	31,167	58.8%	
Private Hospital	9210	17.4%	
University	6193	11.7%	
Medical centre	2359	4.4%	
1st level	2088	3.9%	
Foundation	1869	3.5%	
other	139	0.3%	

^a^: Fisher’s exact test; ^b^: Pearson chi-square test.

**Table 2 children-10-01182-t002:** Annual diagnosis, populations under 18 and annual incidences of CP that occurred within Türkiye during the period spanning from 2016 to 2022.

Years	Annually Diagnosed CP	Population under 18	Incidence (per 1000)
2016	25,634	24,223,372	1.05
2017	24,870	24,224,739	1.03
2018	20,928	24,258,266	0.86
2019	17,762	24,196,448	0.73
2020	10,800	23,979,485	0.45
2021	11,807	23,960,578	0.49
2022	11,268	23,823,305	0.47

**Table 3 children-10-01182-t003:** Demographic data of patients who underwent surgical treatment between 2016 and 2022.

Age	N	%
0–3 years	1910	19.80%
4–6 years	3195	33.10%
7–10 years	2023	20.90%
Older than 10	2530	26.20%
**Gender**		
Male	5673	58.70%
Female	3985	41.30%
**Geographic Region**		
Marmara	2994	31.00%
Central Anatolia	1967	20.40%
Aegean	1770	18.30%
Mediterranean	1243	12.90%
Southeast Anatolia	864	8.90%
East Anatolia	580	6.00%
North Anatolia	240	2.50%
**Citizenship**		
Citizen	9515	98.50%
Immigrant	143	1.50%

**Table 4 children-10-01182-t004:** Orthopaedic surgical procedures by age groups in children with CP.

	0–3 Years	4–6 Years	7–10 Years	Older 10 Years	*n*
Botulinum Toxin Injection	1082 24.6%	1974 44.9%	793 18.1%	543 12.4%	4392
Joint Reconstruction	59 12.9%	78 17.1%	72 15.8%	248 54.3%	457
Tenoplasty/Tenotomy/Myotomy	753 16.5%	1094 24.0%	1097 24.1%	1611 35.4%	4555
Deformity Correction	16 6.3%	49 19.3%	61 24.0%	128 50.4%	254

## Data Availability

Data available on request due to restriction privacy and ethical.

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
