# Peer review of "Prevalence, Incidence, and Surgical Treatment Trends of Cerebral Palsy across Türkiye: A Nationwide Cohort Study"

_children, 2023, doi:10.3390/children10071182_

Round 1
Reviewer 1 Report
1. This is a nationwide cohort study of CP, but the contents detailed the surgical treatment. It is thus advised to rename the title as: "Prevalence, incidence and surgical treatment trends of cerebral palsy in Turkey...". 2. A total of 184311 patients were diagnosed as CP, yet only 53027 children (28.8%) were evaluated. The collection rate was low and representative was challenged, at least, should be mentioned as the weak point of this study. 3. Among 53027 CP patients, 9658 (18.2%) undergone orthopedic surgery. What were the other major group treated? medically, or others? 4. The typing of the references should follow the instruction and template of the journal.
The spellings and orders of affiliations are wrong in some place, such as: 1. Depratment of orthopedics 2. Department of..., ...Hospital, city place, country name
Author Response
We thank you for your thoughtful suggestions and insights. The manuscript has benefited from these insightful suggestions. The manuscript has been rechecked, and the necessary changes have been made following your recommendations. The responses to all comments have been prepared and are given in the Word file.
Q1: This is a nationwide cohort study of CP, but the contents detailed the surgical treatment. It is thus advised to rename the title as: "Prevalence, incidence and surgical treatment trends of cerebral palsy in Turkey...".
Answer: In line with your suggestions, the title of our study was changed to 'Prevalence, Incidence, and Surgical Treatment Trends of Cerebral Palsy Across Turkey: A Nationwide Cohort Study'.
Q2: A total of 184311 patients were diagnosed as CP, yet only 53027 children (28.8%) were evaluated. The collection rate was low and representative was challenged, at least, should be mentioned as the weak point of this study.
Answer: For the purpose of demographic analysis in our research, we incorporated individuals born in the year between 2016 and 2022, as this cohort provided more refined data through the utilization of the e-Nabız system. According to your suggestions, the following sentences has been added to the method section.
‘As of 2016, the e-Nabız registration system data in our country is accessible with the highest degree of accuracy. Hence, we utilized the dataset of individuals born within the timeframe of 2016 to 2022 to conduct incidence and prevalence analyses, and to more precisely assess the demographic characteristics of patients who have been diagnosed with cerebral palsy. However, while giving information about surgical interventions, we evaluated the surgical procedures performed at the same time. To clarify, individuals who received a diagnosis of cerebral palsy prior to 2016 were eligible to participate in the surgical interventions conducted from 2016 to 2022.’
Q3: Among 53027 CP patients, 9658 (18.2%) undergone orthopedic surgery. What were the other major group treated? medically, or others?
Answer: In our nation, individuals diagnosed with cerebral palsy receive comprehensive care through a multidisciplinary approach that involves various medical professionals, including neurosurgeons, spine surgeons, physical medicine specialists, rehabilitation professionals, child development specialists, paediatric neurologists, and orthopaedic surgeons. Nonetheless, the present investigation, which was carried out under the auspices of the Turkish Ministry of Health, was limited to the retrieval of information pertaining solely to orthopaedic surgical procedures administered to individuals. Regrettably, we were unable to obtain data regarding the treatment modalities administered to non-surgically managed patients, due to this limitation.
Q4: The typing of the references should follow the instruction and template of the journal.
Answer: The recommendations provided were taken into consideration and appropriate modifications were implemented accordingly.
Q5: The spellings and orders of affiliations are wrong in some place, such as: 1. Depratment of orthopedics 2. Department of..., ...Hospital, city place, country name
Answer: The recommendations provided were taken into consideration and appropriate modifications were implemented accordingly.

Reviewer 2 Report
This is an important study, providing the first description of the total population of children with CP in Turkey, describing orthopaedic surgical procedures and making the case for the need of a national registry.
Title - I suggest adding 'surgical' to clarify that it is surgical treatment trends that you will be discussing - not other forms of treatment.
The manuscript would benefit from additional information and clarification for the reader.
Sample - I am somewhat confused about your sample. Your cohort included 53,027 children born within the years 2016 and 2022. 9658 children in the study cohort had at least one surgery performed between 2016 and 2022. Children born between 2016 and 2022 can only be a maximum of 6 years old. The surgical analysis includes children aged 7-10 and over 10 years. Are there two study cohorts - one used for the prevalence/incidence calculations and another for the surgical analysis? This requires clarification.
Geographical regions - the reader, unfamiliar with Turkey, would benefit from some orientation to the different regions to understand the data presented. For example, are some regions more rural/poorer, do some regions have more health facilities, while in other regions individuals have to travel to a different region for services?
I also noted differences in how the regions are named in Table 1 vs 2 - I assume this is a formatting error - but there are 7 lines of data in Table 1 and only 5 region names - which appear to be combined - in comparison to Table 2.
The prevalence of CP was calculated to be 0.9 per 1000 population of children under 18 - and was stable in the period from 2016 to 2022. Incidence was challenging to calculate because of increasing numbers of immigrants. This is unclear - surely immigrants with CP would add to the prevalence rather than the incidence?
Figure 1 shows incidence of CP from 2016 to 2022 - the actual numbers drop considerably. Has the birth rate also dropped substantially or has medical intervention changed?
Static prevalence may not indicate high mortality if the birth rate or incidence is dropping? Prevalence of CP would be expected to increase, if most children survive - and particularly if immigration is adding to the numbers of children with CP - this needs to more clarification in the paper.
GMFCS - why is no data provided on GMFCS level? If not available, this should be explained - and is a limitation of the paper. Even providing information on topography - quadriplegia, diplegia, hemiplegia (which it appears you have as the ICD codes are recorded) would give some idea of functional and severity profile.
Age bands - I agree that it is useful to stratify according to age - but the divisions chosen should be justified in the methods.
Categories of surgical interventions - more details on what interventions are included in each category is needed. Surgical interventions differ based on GMFCS level. It would also be helpful to know if upper or lower limbs were targeted for the different procedures. While the overall data is useful, more details would be useful. In the discussion, you note that other authors found no statistically significant associations between age, sex, an anatomical location and surgeries - but you report only the most common surgical interventions - and that these were so, regardless of anatomical location. Did you perform statistical analyses? It would be helpful to have more descriptive detail, and statistical analysis if appropriate.
Statistical analyses - actual numbers and percentages are primarily reported. The only mean and standard deviation I noted was age of surgical intervention - and that is in the discussion, not results. I did not see results for the other statistical analyses mentioned in the methods section. Table 1 notes a significance value - what analysis was conducted? This should be noted in a footnote.
The authors are to be commended for their intention to develop a registry for all individuals with CP and make a number of excellent recommendations for the data that should be included - and that would be very relevant for future studies and analyses.
Author Response
We thank you for your thoughtful suggestions and insights. The manuscript has benefited from these insightful suggestions. The manuscript has been rechecked, and the necessary changes have been made following your recommendations. The responses to all comments have been prepared and are given in the Word file.
Q1: Title - I suggest adding 'surgical' to clarify that it is surgical treatment trends that you will be discussing - not other forms of treatment.
Answer: In line with your suggestions, the title of our study was changed to 'Prevalence, Incidence, and Surgical Treatment Trends of Cerebral Palsy Across Turkey: A Nationwide Cohort Study'.
Q2: The manuscript would benefit from additional information and clarification for the reader.
Sample: Sample - I am somewhat confused about your sample. Your cohort included 53,027 children born within the years 2016 and 2022. 9658 children in the study cohort had at least one surgery performed between 2016 and 2022. Children born between 2016 and 2022 can only be a maximum of 6 years old. The surgical analysis includes children aged 7-10 and over 10 years. Are there two study cohorts - one used for the prevalence/incidence calculations and another for the surgical analysis? This requires clarification.
Answer: According to your suggestions, the following sentences has been added to the method section.
‘As of 2016, the e-Nabız registration system data in our country is accessible with the highest degree of accuracy. Hence, we utilized the dataset of individuals born within the timeframe of 2016 to 2022 to conduct incidence and prevalence analyses, and to more precisely assess the demographic characteristics of patients who have been diagnosed with cerebral palsy. However, while giving information about surgical interventions, we evaluated the surgical procedures performed at the same time. To clarify, individuals who received a diagnosis of cerebral palsy prior to 2016 were eligible to participate in the surgical interventions conducted from 2016 to 2022.’
Geographical regions - the reader, unfamiliar with Turkey, would benefit from some orientation to the different regions to understand the data presented. For example, are some regions more rural/poorer, do some regions have more health facilities, while in other regions individuals have to travel to a different region for services?
Answer: Thank you for your suggestions. The following sentences have been added to the method part.
‘Turkey is comprised of seven distinct geographical regions, namely Marmara, Central Anatolia, Aegean, Mediterranean, Southeast Anatolia, Eastern Anatolia, and Northern Anatolia. These regions exhibit variations in several dimensions, including socio-economic advancement and availability of healthcare facilities. The urban areas of the nation, which serve as significant hubs for industry and commerce, are situated in the Marmara, Aegean, Central Anatolia, and Mediterranean regions. In contrast, the rural regions of Eastern Anatolia, Southeast Anatolia, and Northern Anatolia are predominantly agricultural. Due to this rationale, while primary healthcare services are available in Turkey's Eastern, Northern, and Southeast regions, facilities in other areas are favoured for complex diagnoses and treatments of various health conditions, such as cerebral palsy.’
I also noted differences in how the regions are named in Table 1 vs 2 - I assume this is a formatting error - but there are 7 lines of data in Table 1 and only 5 region names - which appear to be combined - in comparison to Table 2.
Answer: Thank you for bringing this to my attention. The technical error on the table was rectified, thereby resolving the error that had occurred.
|
|
N |
% |
p |
|
Gender Male Female |
29606 23421 |
55.8% 44.2% |
p˂0.001 |
|
Geographic Region Southeast Anatolia Marmara Mediterranean Central Anatolia East Anatolia Aegean North Anatolia |
14010 13816 7752 6456 4062 4027 2903 |
26.4% 26.1% 14.6% 12.2% 7.7% 7.6% 5.5% |
p˂0.001 |
|
Healthcare Provider 2nd/3rd levels affiliated. Private Hospital University Medical center 1st level Foundation other |
31167 9210 6193 2359 2088 1869 139 |
58.8% 17.4% 11.7% 4.4% 3.9% 3.5% .3% |
p˂0.001 |
The prevalence of CP was calculated to be 0.9 per 1000 population of children under 18 - and was stable in the period from 2016 to 2022. Incidence was challenging to calculate because of increasing numbers of immigrants. This is unclear - surely immigrants with CP would add to the prevalence rather than the incidence?
Figure 1 shows incidence of CP from 2016 to 2022 - the actual numbers drop considerably. Has the birth rate also dropped substantially or has medical intervention changed?
Static prevalence may not indicate high mortality if the birth rate or incidence is dropping? Prevalence of CP would be expected to increase, if most children survive - and particularly if immigration is adding to the numbers of children with CP - this needs to more clarification in the paper.
Answer: Following paragraph also was given in the discussion section.
The incidence of a disease is a metric that quantifies the frequency of its manifestation, while prevalence is a metric that quantifies its overall presence. The incidence metric identifies new cases, whereas the prevalence metric provides information on new and pre-existing cases. From this perspective, it is believed that while both data sets are impacted by migration, the incidence prevalence is likely to be more significantly affected. Since the inception of the e-Nabız patient registration system in 2016, the incidence data has been consistently recorded with high accuracy. Notably, the incidence rate was observed to be high during the years 2016 and 2017. However, in the years 2020 and 2021, the incidence rate has declined, which can be attributed to the limited number of hospital admissions resulting from the COVID-19 pandemic. This is thought to be the primary factor contributing to the reduction in occurrence.
GMFCS - why is no data provided on GMFCS level? If not available, this should be explained - and is a limitation of the paper. Even providing information on topography - quadriplegia, diplegia, hemiplegia (which it appears you have as the ICD codes are recorded) would give some idea of functional and severity profile.
Answer: Following sentences added into the limitations section, located in the final paragraph of the Discussion section.
‘Even though GMFCS scores are documented within the health institution's data of the patient's follow-up in our country, it is regrettable that there is a lack of a distinct data tab within the e-Nabız system, which would enable us to record this information for the entire national cohort that we evaluate. Consequently, the GMFCS scores of the patients within our study cohort needed to be more attainable.’
‘The implementation of a nationwide tracking system that incorporates comprehensive demographic information, socio-economic indicators, GMFCS scores, the anatomical regions targeted by surgical interventions, the various subgroups of surgical interventions, and any complications would be highly advantageous for healthcare providers and individuals with cerebral palsy in devising appropriate treatment plans and follow-up protocols.’
Age bands - I agree that it is useful to stratify according to age - but the divisions chosen should be justified in the methods.
Answer: Following sentences added into material and methods section according to your suggestions.
‘The primary rationale behind categorizing the patients based on their age pertains to the unavailability of the GMFCS scores of the patients. The grouping of patients based on age can be informative in assessing surgical interventions, given that those with a higher GMFCS score tend to require such procedures at an earlier stage.’
Categories of surgical interventions - more details on what interventions are included in each category is needed. Surgical interventions differ based on GMFCS level. It would also be helpful to know if upper or lower limbs were targeted for the different procedures. While the overall data is useful, more details would be useful. In the discussion, you note that other authors found no statistically significant associations between age, sex, an anatomical location and surgeries - but you report only the most common surgical interventions - and that these were so, regardless of anatomical location. Did you perform statistical analyses? It would be helpful to have more descriptive detail, and statistical analysis if appropriate.
Answer: Following sentences added into the limitations section, located in the final paragraph of the Discussion section.
‘Despite its significance in patient follow-up, the e-Nabız system is currently in its nascent stage with regards to the follow-up of individuals with cerebral palsy. Regrettably, the current documentation remains inadequate in capturing crucial data points such as the GMFCS scores of patients, the anatomical regions targeted by surgical interventions, the various subgroups of surgical interventions, and any complications that may arise from the use of the e-Nabız system.’
‘The implementation of a nationwide tracking system that incorporates comprehensive demographic information, socio-economic indicators, GMFCS scores, the anatomical regions targeted by surgical interventions, the various subgroups of surgical interventions, and any complications would be highly advantageous for healthcare providers and individuals with cerebral palsy in devising appropriate treatment plans and follow-up protocols.’
Statistical analyses - actual numbers and percentages are primarily reported. The only mean and standard deviation I noted was age of surgical intervention - and that is in the discussion, not results. I did not see results for the other statistical analyses mentioned in the methods section. Table 1 notes a significance value - what analysis was conducted? This should be noted in a footnote.
Answer: Footnotes regarding the type of statistical analyses were given under the Table 1.

Reviewer 3 Report
This paper reports the prevalence of cerebral palsy (CP) in Turkish children and how many had different types of surgery. The reporting is clear and transparent. The English expression is excellent. Although I have a number of recommendations for this paper (particularly in the Discussion section), I think that they can all be achieved.
1. As this is an international journal, could the authors please give a little bit of information about the process of diagnosis in their country? Is there any data about age of diagnosis? Do you think that all children with CP are being identified or only the more severe ones? Are there influences in Turkish society that make it more or less likely that children will be diagnosed (e.g., funding opportunities, social stigma)? This would go into the Introduction.
2. The aim is given as: “This study aims to identify the demographics of CP cases among Turkish children, the clinical associations, and surgical preferences.” What is meant by “clinical associations”?
3. Change “European, Australian, and American countries” to “Europe, Australia, and North America”. (Australia is only one country and most of the American data are from the US and Canada, not South American countries.)
4. How did you arrive at the sample of 53027 children? You say that “during the period from 2016 to 2022, a total of 184311 patients were diagnosed with CP.” But “the present study analysed the demographic information of a sample of 53027 children born within the years 2016-2022.” Why didn’t you analyse the full 184311? (In fact, I would expect even more than 184311, because there were 184311 newly diagnosed in 2016-2022, but there were also the ones who had already received their diagnosis but were still aged <18. So perhaps I am misunderstanding something here. Could you please clarify?)
5. In Table 1, there appear to be a couple of geographic regions missing, because the heading “Geographic Region” has an N of 14010 and a prevalence of 26.4%, and because there is an extra row below any geographic region.
6. You say that “In 2022, an estimated 23823305 children under the age of 18 lived in Türkiye.” Could you please give the number of children with CP under the age of 18? It is evidently not the figure of 184311 already given because 184311/23823305 x 1000 = 7.74, not 0.9. To get a figure of 0.9, it must have been something like 21441, but that figure hasn’t been given anywhere that I can see.
7. Can you provide any data on severity of CP? E.g., GMFCS, or how many could walk, or how many were verbal? This would give some idea of how many of the children with milder CP are being diagnosed, and would also allow comparison of your data with the data from other countries.
8. Please indicate whether the incidences given in Figure 1 are per hundred or per thousand.
9. Is it correct to call “botulinum toxin injections” a type of surgery?
10. Please list the surgeries covered under each of the surgical categories, either in a table or in text.
11. Is it possible to provide any breakdown (e.g., by area of surgery or other description of surgery), as some of these categories encompass a variety of surgeries?
12. Tables 2 and 3 show the percentages of children who underwent surgery, but some children had more than one surgery. Could numbers of children with multiple surgeries also be indicated, and how many they had? Some procedures would be expected to occur only once. Botulinum toxin injections occur repeatedly, so may be under-represented in your data if you give only the number of children who received that type of procedure at least once.
13. The Discussion includes several paragraphs that give very general information, not a specific discussion of findings and not comparison of findings with previous research. This includes: the first half of paragraph 2; paragraphs 3 and 4 (which would not be out of place in the Introduction), and the first few sentences of paragraph 5. Generally speaking, material doesn’t belong in the Discussion section, if it could have been written before seeing the results. The Discussion should reflect on the results.
14. In the Discussion section, please compare the prevalence and incidence of CP found in the present study with those of other studies. I think yours is much lower. Why is that? Is it a true reflection of the prevalence of CP in your country? Or are there some children with CP not being diagnosed, and who and why? Or are the data incomplete and are doctors not recording all the diagnoses?
15. Please consider discussing the immigrants more. Are these children who themselves have immigrated? Or does it include children born in Türkiye, but whose parents immigrated before they were born? Does their lower rate of surgeries reflect something about funding available for people born in the country versus immigrants? Are there any comparative socioeconomic data for this group that might explain differences observed?
16. Could you please discuss the significance of the regional comparisons for the benefit of non-Turkish readers who do not know which areas are high or low income or which areas are more urban or rural?
17. It is good that limitations are discussed. But would it be possible to address some of these limitations? For example, one of the limitations mentioned is missing or incomplete data. How much data were missing? Could this be reported in the Results section?
18. Another limitation mentioned is lack of socioeconomic data. However, you know something about the regions and the immigrant groups in your study, so could you report (non-CP) Turkish population data about these subgroups of the Turkish population to help you interpret your data?
19. Please explain more about the “lack of a standard definition and diagnostic criteria for CP across different hospitals”. What do you know about the different definitions used in different hospitals?
20. The conclusion mentions for the first time “the impending establishment of a national registry system for CP in our country”. This is good news. But the conclusion to this paper is not the place to say this, because it is not a conclusion from the findings of this study. (This also applies to the conclusion of the Abstract.) If you want, you could mention in the Introduction that the present study is based on records from the Turkish Ministry of Health because there is no Turkish registry as yet, although one is soon to be established.
English is good.
Author Response
We thank you for your thoughtful suggestions and insights. The manuscript has benefited from these insightful suggestions. The manuscript has been rechecked, and the necessary changes have been made following your recommendations. The responses to all comments have been prepared and are given in the Word file.
Q1: As this is an international journal, could the authors please give a little bit of information about the process of diagnosis in their country? Is there any data about age of diagnosis? Do you think that all children with CP are being identified or only the more severe ones? Are there influences in Turkish society that make it more or less likely that children will be diagnosed (e.g., funding opportunities, social stigma)? This would go into the Introduction.
Answer: Following paragraph added in the introduction section according to your suggestions
‘A combination of delayed motor milestones, abnormal neurologic examination, the persistence of primitive reflexes, and abnormal postural reactions are indicative of a potential diagnosis of CP. Besides, the neonatal medical history can potentially elicit concerns and requests for heightened monitoring. (ATIF TANI). Sequential assessments are crucial in tracking advancements in cases where cerebral palsy is suspected. The identification of brain damage is typically simplified when it is supported by documented evidence from cranial ultrasound, computed tomography (CAT scan), or magnetic resonance imaging. The amalgamation of natal history, risk factors, and physical examination findings determines Türkiye's diagnostic requirements for CP. Typically, children with more severe symptoms are often diagnosed by the age of six months, while milder cases may require a longer period for diagnosis.’
- The aim is given as: “This study aims to identify the demographics of CP cases among Turkish children, the clinical associations, and surgical preferences.” What is meant by “clinical associations”?
Answer: The phrase "clinical associations", which caused confusion, was removed.
- Change “European, Australian, and American countries” to “Europe, Australia, and North America”. (Australia is only one country and most of the American data are from the US and Canada, not South American countries.)
Answer: Your suggested corrections have been made.
- How did you arrive at the sample of 53027 children? You say that “during the period from 2016 to 2022, a total of 184311 patients were diagnosed with CP.” But “the present study analysed the demographic information of a sample of 53027 children born within the years 2016-2022.” Why didn’t you analyse the full 184311? (In fact, I would expect even more than 184311, because there were 184311 newly diagnosed in 2016-2022, but there were also the ones who had already received their diagnosis but were still aged <18. So perhaps I am misunderstanding something here. Could you please clarify?)
Answer: According to your suggestions, the following sentences has been added to the method section.
‘During the period spanning from 2016 to 2022, a cumulative sum of 184311 patients diagnosed with CP was identified, and their respective cases were documented within the e-Nabız system. Among the patient population, 53027 individuals were identified who were born within the time frame of 2016 to 2022.’
‘As of 2016, the e-Nabız registration system data in our country is accessible with the highest degree of accuracy. Hence, we utilized the dataset of individuals born within the timeframe of 2016 to 2022 to conduct incidence and prevalence analyses, and to more precisely assess the demographic characteristics of patients who have been diagnosed with cerebral palsy. However, while giving information about surgical interventions, we evaluated the surgical procedures performed at the same time. To clarify, individuals who received a diagnosis of cerebral palsy prior to 2016 were eligible to participate in the surgical interventions conducted from 2016 to 2022.’
- In Table 1, there appear to be a couple of geographic regions missing, because the heading “Geographic Region” has an N of 14010 and a prevalence of 26.4%, and because there is an extra row below any geographic region.
Answer: Thank you for bringing this to our attention. The technical error on the table was rectified, thereby resolving the error that had occurred.
- You say that “In 2022, an estimated 23823305 children under the age of 18 lived in Türkiye.” Could you please give the number of children with CP under the age of 18? It is evidently not the figure of 184311 already given because 184311/23823305 x 1000 = 7.74, not 0.9. To get a figure of 0.9, it must have been something like 21441, but that figure hasn’t been given anywhere that I can see.
Answer: Thank you for bringing this to our attention. After your warnings, we realized that we made a mistake in our incidence and prevalence calculations. We reviewed our data with professional support and made our calculations again.
As of the year 2022, the number of individuals who have been diagnosed with CP and are under the age of 18 in our nation is 184311. The aggregate count of individuals within our populace who are below the age of 18 is 23823305. As per your information, the incidence rate among children under 18 is 7.74 per 1000. Thank you for your warnings and contributions.
- Can you provide any data on severity of CP? E.g., GMFCS, or how many could walk, or how many were verbal? This would give some idea of how many of the children with milder CP are being diagnosed, and would also allow comparison of your data with the data from other countries.
Answer: Despite the existence of local registry systems in healthcare institutions in our country that document data such as physical examination findings and GMFCS scores of patients, these data are not integrated with the e-Nabız system, thereby impeding access to such information. The aforementioned circumstance, which constitutes a primary constraint of our investigation, has been duly acknowledged in the limitations section of our report. Furthermore, the implementation of an electronic diagnostic and follow-up system tailored for cerebral palsy is an unavoidable necessity within our nation.
‘Even though GMFCS scores are documented within the health institution's data of the patient's follow-up in our country, it is regrettable that there is a lack of a distinct data tab within the e-Nabız system, which would enable us to record this information for the entire national cohort that we evaluate. Consequently, the GMFCS scores of the patients within our study cohort needed to be more attainable.’
Despite its significance in patient follow-up, the e-Nabız system is currently in its nascent stage with regards to the follow-up of individuals with cerebral palsy. Regrettably, the current documentation remains inadequate in capturing crucial data points such as the GMFCS scores of patients, the anatomical regions targeted by surgical interventions, the various subgroups of surgical interventions, and any complications that may arise from the use of the e-Nabız system.’
‘The implementation of a nationwide tracking system that incorporates comprehensive demographic information, socio-economic indicators, GMFCS scores, the anatomical regions targeted by surgical interventions, the various subgroups of surgical interventions, and any complications would be highly advantageous for healthcare providers and individuals with cerebral palsy in devising appropriate treatment plans and follow-up protocols.
- Please indicate whether the incidences given in Figure 1 are per hundred or per thousand.
Thank you for bringing this to our attention. After your warnings, we realized that we made a mistake in our incidence and prevalence calculations. We reviewed our data with professional support and made our calculations again.
We recalculated our annual incidences between the years 2016-2022 and included the incidences of every 10000 individuals under the age of 18 in the figure with the new Figure1.
- Is it correct to call “botulinum toxin injections” a type of surgery?
Answer: To determine the population of patients who had undergone surgical interventions, the Turkish Ministry of Health codes known to be applied for patients with CP were selected (https://skrs.saglik.gov.tr/). The procedures denoted by codes 611540 and 611550 pertain to the injection of botulinum toxin and are classified as surgical interventions. Although this procedure is usually performed by orthopedic surgeons in operating room conditions in our country, it is within our knowledge that it can also be applied in outpatient clinic conditions.
- Please list the surgeries covered under each of the surgical categories, either in a table or in text.
- Is it possible to provide any breakdown (e.g., by area of surgery or other description of surgery), as some of these categories encompass a variety of surgeries?
Answer: In the method section, brief information about the subgroups of surgical interventions was given. The table below contains the codes of the surgical interventions applied to the patients in our study. We share this table with you, but since we think that it will complicate our study, we consider it more appropriate to report surgical interventions under four main headings: surgical interventions such as botulinum toxin injection, joint reconstruction (arthrodesis, surgical reduction of joint dislocations, surgical loosening of joint contractions, etc.), tenoplasty-myoplasty-tendon transfer. and deformity correction (distal femoral extension osteotomies, proximal femur derotation osteotomies, medial column corrections of the foot, etc.).
|
|
Surgical intervention codes defined for cerebral palsy patients |
|
611.540 |
Botulinum toxin injection, profound |
|
611.550 |
Botulinum toxin injection, superficial |
|
611.760 |
Conracture relaese, large joints |
|
611.770 |
Conracture relaese, middle joints |
|
611.780 |
Conracture relaese, minor joints |
|
611.980 |
Tendon transfer, single |
|
611.990 |
Tendon transfer, multiple |
|
612.000 |
Tenodesis |
|
612.010 |
Tenolysis |
|
612.020 |
Tenoplasty, myoplasty, fascia release, single |
|
612.030 |
Tenoplasty, myoplasty, fascia release, multiple |
|
612.040 |
Tenotomy, myotomy |
|
612.150 |
Reconstriction of large joint dislocation |
|
612.160 |
Reconstriction of middle joint dislocation |
|
612.170 |
Reconstriction of minor joint dislocation |
|
612.590 |
Arthrodesis of large joint |
|
612.600 |
Arthrodesis of middle joint |
|
612.610 |
Arthrodesis of minor joint |
|
613.460 |
Osteoclasia |
|
614.290 |
Releasing the large joint contracture with an external fixator |
|
614.300 |
Releasing the middle joint contracture with an external fixator |
|
614.320 |
Large bone lengthening/Deformity correction surgery |
|
614.330 |
Middle bone lengthening/Deformity correction surgery |
|
614.340 |
Small bone lengthening/Deformity correction surgery |
- Tables 2 and 3 show the percentages of children who underwent surgery, but some children had more than one surgery. Could numbers of children with multiple surgeries also be indicated, and how many they had? Some procedures would be expected to occur only once. Botulinum toxin injections occur repeatedly, so may be under-represented in your data if you give only the number of children who received that type of procedure at least once.
Answer: Following sentences added in the methods section.
‘The study assessed the most invasive surgical procedures conducted on pediatric patients who underwent multiple surgical interventions. To clarify, the documentation of patients who received repetitive botulinum toxin injections from 2016 to 2022 was categorized as botulinum toxin injection interventions. However, if a patient underwent a procedure such as a deformity correction or osteotomy during the same time frame, the invasive procedure was incorporated into the surgical assessment.’
- The Discussion includes several paragraphs that give very general information, not a specific discussion of findings and not comparison of findings with previous research. This includes: the first half of paragraph 2; paragraphs 3 and 4 (which would not be out of place in the Introduction), and the first few sentences of paragraph 5. Generally speaking, material doesn’t belong in the Discussion section, if it could have been written before seeing the results. The Discussion should reflect on the results.
Answer: We concur with your perspective on this topic and acknowledge that we have presented introductory information in the discussion section. Nevertheless, the prevailing publication policies of the journal require a comprehensive discourse within a minimum 4500-word limit. Thus, the aforementioned paragraphs have been appropriately incorporated into this particular section.
- In the Discussion section, please compare the prevalence and incidence of CP found in the present study with those of other studies. I think yours is much lower. Why is that? Is it a true reflection of the prevalence of CP in your country? Or are there some children with CP not being diagnosed, and who and why? Or are the data incomplete and are doctors not recording all the diagnoses?
Answer: Thank you for bringing this to our attention. After your warnings, we realized that we made a mistake in our incidence and prevalence calculations. We reviewed our data with professional support and made our calculations again.
As of the year 2022, the number of individuals who have been diagnosed with CP and are under the age of 18 in our nation is 184311. The aggregate count of individuals within our populace who are below the age of 18 is 23823305. As per your information, the incidence rate among children under 18 is 7.74 per 1000. Thank you for your warnings and contributions.
Furthermore, following sentences added in the discussion section.’ The incidence of a disease is a metric that quantifies the frequency of its manifestation, while prevalence is a metric that quantifies its overall presence. The incidence metric identifies new cases, whereas the prevalence metric provides information on new and pre-existing cases. From this perspective, it is believed that while both data sets are impacted by migration, the incidence prevalence is likely to be more significantly affected. Since the inception of the e-Nabız patient registration system in 2016, the incidence data has been consistently recorded with high accuracy. Notably, the incidence rate was observed to be high during the years 2016 and 2017. However, in the years 2020 and 2021, the incidence rate has declined, which can be attributed to the limited number of hospital admissions resulting from the COVID-19 pandemic. This is thought to be the primary factor contributing to the reduction in occurrence.’
- Please consider discussing the immigrants more. Are these children who themselves have immigrated? Or does it include children born in Türkiye, but whose parents immigrated before they were born? Does their lower rate of surgeries reflect something about funding available for people born in the country versus immigrants? Are there any comparative socioeconomic data for this group that might explain differences observed?
In our study group, we included both immigrant children who immigrated to our country and those who were born in our country after their families migrated to our country.The influx of refugees to our nation experienced an escalation subsequent to 2012. From this year, the national health system has assigned a registration number to legally recognized asylum seekers, enabling them to access free healthcare services. Regrettably, the extent to which these children can benefit from health services remains uncertain due to various social challenges and impediments, including linguistic barriers. Unfortunately, we do not have data on the socioeconomic differences of this group.
- Could you please discuss the significance of the regional comparisons for the benefit of non-Turkish readers who do not know which areas are high or low income or which areas are more urban or rural?
Answer: Thank you for your suggestions. The following sentences have been added to the method part.
‘Türkiye is comprised of seven distinct geographical regions, namely Marmara, Central Anatolia, Aegean, Mediterranean, Southeast Anatolia, Eastern Anatolia, and Northern Anatolia. These regions exhibit variations in several dimensions, including socio-economic advancement and availability of healthcare facilities. The urban areas of the nation, which serve as significant hubs for industry and commerce, are situated in the Marmara, Aegean, Central Anatolia, and Mediterranean regions. In contrast, the rural regions of Eastern Anatolia, Southeast Anatolia, and Northern Anatolia are predominantly agricultural. Due to this rationale, while primary healthcare services are available in Turkey's Eastern, Northern, and Southeast regions, facilities in other areas are favoured for complex diagnoses and treatments of various health conditions, such as cerebral palsy.’
- It is good that limitations are discussed. But would it be possible to address some of these limitations? For example, one of the limitations mentioned is missing or incomplete data. How much data were missing? Could this be reported in the Results section?
Answer: In our study, we have a data set containing dozens of data from approximately 24 million children. We think that it is not very possible to specify all of them separately as data loss. Currently, this is a common problem of retrospective studies with large cohorts.
- Another limitation mentioned is lack of socioeconomic data. However, you know something about the regions and the immigrant groups in your study, so could you report (non-CP) Turkish population data about these subgroups of the Turkish population to help you interpret your data?
Answer: We have added general information about the socio-economic level of geographical regions to the method section following:
‘Türkiye is comprised of seven distinct geographical regions, namely Marmara, Central Anatolia, Aegean, Mediterranean, Southeast Anatolia, Eastern Anatolia, and Northern Anatolia. These regions exhibit variations in several dimensions, including socio-economic advancement and availability of healthcare facilities. The urban areas of the nation, which serve as significant hubs for industry and commerce, are situated in the Marmara, Aegean, Central Anatolia, and Mediterranean regions. In contrast, the rural regions of Eastern Anatolia, Southeast Anatolia, and Northern Anatolia are predominantly agricultural. Due to this rationale, while primary healthcare services are available in Turkey's Eastern, Northern, and Southeast regions, facilities in other areas are favoured for complex diagnoses and treatments of various health conditions, such as cerebral palsy.’
However, we are aware that this evaluation will not be sufficient to evaluate individual characteristics.
- Please explain more about the “lack of a standard definition and diagnostic criteria for CP across different hospitals”. What do you know about the different definitions used in different hospitals?
Answer: Following sentences added in Limitations
‘The primary factor contributing to this scenario is physicians' varying expertise levels across different healthcare facilities in diagnosing cerebral palsy. Less experienced physicians may struggle to identify subtle clinical presentations of the condition.’
- The conclusion mentions for the first time “the impending establishment of a national registry system for CP in our country”. This is good news. But the conclusion to this paper is not the place to say this, because it is not a conclusion from the findings of this study. (This also applies to the conclusion of the Abstract.) If you want, you could mention in the Introduction that the present study is based on records from the Turkish Ministry of Health because there is no Turkish registry as yet, although one is soon to be established.
Answer: Following paragraph added in the introduction section according to your suggestions
‘In our study, the prevalence of cerebral palsy in children under the age of 18 was found to be 7.74/1000. The minimum and maximum incidence rates of cerebral palsy among children between 2016 and 2022 were calculated to be 4 and 10 per 10000, respectively.The majority of the surgical procedures performed are tenoplasty-myoplasty- tendon transfer operations (47.1%) and botulinum toxin injections (46.5%). Although the centres in which patients are diagnosed has shown a uniform distribution throughout our country, surgical procedures are often carried out in urban regions.’
In addition, we believe in the necessity of a cerebral palsy registration system as authors in order to continue to recognize the patients with standardized diagnostic methods of cerebral palsy patients in our country and to emphasize this situation, which is an important public health problem, in Conclision section, which is an important public health problem. We appreciate your comprehension of the present circumstances.

Reviewer 4 Report
Q1: line 17: Our aim “is”to identify the demographics of CP in Turkish children in addition to clinical associations and surgical preferences.
Q2: line 20 : underwent orthopaedic surgery------ underwent at least(?) one orthopaedic surgery
Q3: line 48: there are limited data from other parts of the world.
Q4: Discussion: line 156-158 The study revealed notable variations in the geographic distribution of medical facilities where diagnoses of CP in children were made and where surgical interventions were carried out.
You need to explain why.
Q5: line 192-204
You cited references 18-21 to support your findings pertaining to the table 3. This is very good. However, do you think that the way of your explanations and interpretation of the data were a little bit “simple”? For example, while you are in the moment for the consideration of taking an operation in despite of its advantages, most people may step back and think more and may possibly delay the procedure and wait for a “better one”, or take a relatively “non-invasive”. This is a human-nature.
Would it be possible that it is the human-nature that led to your results? You should consider this possibility in our discussion. Although your data show the noninvasive procedures, such as BTX injection is much more commonly than others, it seems that your findings cannot go in line with the reference 19, because your data cannot prove that whether the life of quality of these children indeed improved or not after receiving these invasive or noninvasive procedures.
minor English errors
Author Response
We thank you for your thoughtful suggestions and insights. The manuscript has benefited from these insightful suggestions. The manuscript has been rechecked, and the necessary changes have been made following your recommendations. The responses to all comments have been prepared and are given in the Word file.
Q1: line 17: Our aim “is”to identify the demographics of CP in Turkish children in addition to clinical associations and surgical preferences.
Answer: The recommendations provided were taken into consideration and appropriate modifications were implemented accordingly.
‘Our aim is to identify the demographics of CP in Turkish children in addition to clinical associations and surgical preferences.’
Q2: line 20 : underwent orthopaedic surgery------ underwent at least(?) one orthopaedic surgery.
Answer: The terminology was employed in order to delineate the count of patients who underwent surgery, rather than the count of surgical interventions, in the context of our investigation. Furthermore, the inclusion of patients who underwent multiple surgical interventions was also a factor in our study. Following sentences added into the methods section.
‘The study assessed the most invasive surgical procedures conducted on paediatric patients who underwent multiple surgical interventions. To clarify, the documentation of patients who received repetitive botulinum toxin injections from 2016 to 2022 was categorized as botulinum toxin injection interventions. However, if a patient underwent a procedure such as a deformity correction or osteotomy during the same time frame, the invasive procedure was incorporated into the surgical assessment.’
Q3: line 48: there are limited data from other parts of the world.
Answer: The sentence removed from paragraph.
Q4: Discussion: line 156-158 The study revealed notable variations in the geographic distribution of medical facilities where diagnoses of CP in children were made and where surgical interventions were carried out. You need to explain why.
Answer: Following sentences added into the discussion section
‘The primary factor contributing to this situation is the uneven distribution of medical resources across urban and rural areas in our country. While there are no restrictions on diagnosing cerebral palsy in rural regions, surgical interventions and subsequent diagnostic procedures are predominantly accessible in urban centers with more advanced medical infrastructure.’
Q5: line 192-204
You cited references 18-21 to support your findings pertaining to the table 3. This is very good. However, do you think that the way of your explanations and interpretation of the data were a little bit “simple”? For example, while you are in the moment for the consideration of taking an operation in despite of its advantages, most people may step back and think more and may possibly delay the procedure and wait for a “better one”, or take a relatively “non-invasive”. This is a human-nature.
Answer: We agree with you on this. It's in human nature. There is also a shift towards minimally invasive methods in many fields of medicine in the world. But our aim is not to mislead people while giving this information. It would be correct to conclude this paragraph with the following sentence: "The reason why minimally invasive methods and BTX injections are used at young ages is to prevent more invasive and difficult surgical procedures if possible, and to delay them if it is not possible. However, in some patients, unfortunately, it is not even possible to delay major surgical interventions."
Would it be possible that it is the human-nature that led to your results? You should consider this possibility in our discussion. Although your data show the noninvasive procedures, such as BTX injection is much more commonly than others, it seems that your findings cannot go in line with the reference 19, because your data cannot prove that whether the life of quality of these children indeed improved or not after receiving these invasive or noninvasive procedures.
Answer: Following sentence added to paragraph
‘Regrettably, our study was unable to assess the functional outcomes of individuals with cerebral palsy who received surgical intervention.’

Reviewer 5 Report
The purpose of this manuscript is to is manuscript is to assess the prevalence of CP and surgical treatment trends in Turkey. Overall, I believe this important information to present. However, in the current form I have many comments regarding this manuscript.
Lines 43-44 state: “Understanding the characteristics and prevalence of CP is crucial for its management and prevention”.
While I agree with the importance of understanding prevalence and characteristics are important for managing care, I am not sure how this information would affect prevention of CP from occurring. Please consider rewording this sentence or clarify how this information could affect prevention.
Lines 52-53 state: “The body of literature addressing the comprehensive landscape of orthopaedic surgical procedures 53 performed for children with CP is limited.”
I do not agree with this statement, please consider rewording or removing. Why, a quick good scholar search with the terms cerebral palsy and orthopedic surgery in English reports over 4,000 results for 2022 and 2023.
Lines 69-74: How is Turkish Ministry of Health’s database not part of a registry…possibly expound on this database and explain what it is and is not and why a separate CP registry is needed.
Lines: 85-86. Please expand on what criteria were used to decide, “lacking sufficient data”.
Line 87-91. Please clarify and re-write this paragraph. First sentence reports 184311 patients were diagnosed with CP.
Please clarify were these new diagnosis or already diagnosed.
The next sentence says between 2016-2022 53,027 children were born and 9658 had surgery.
This paragraph needs to be re-written to clarify how you identified: 1) how many children born between 2016 and 2022 were diagnosed with CP, 2) at what age were they diagnosed CP, 3) did all of the 9658 children that underwent surgery also previously have a diagnosis of CP?
Perhaps some type of flow diagram would be beneficial.
Lines 105-6: please clarify what ‘deformity correction’ surgical interventions are.
Table 1:
What does the p value in the third column indicate?
How many of these children born were diagnosed with CP between 2016-2022?
Figure 1: please explain what vertical axes represent. Is this a percentage?
It is still unclear how many persons were newly diagnosed with CP between 2016-2022.
If there were a lot of refugees included during this time, then it is still unclear how many persons were newly diagnosed with CP versus persons already diagnosed with CP and immigrated to Turkey.
This information would greatly affect prevalence of CP and must be accounted for in some manner.
Lines 135-143 and Table 2. Please clarify the 9658 patients, all were previously diagnosed with CP?
Also, if 1.5 % of these patients are immigrants, then it seems to me these could be excluded if you are trying to say something about prevalence of CP.
Lines 154-5 state “In this study, we aimed to evaluate the prevalence and incidence of CP in a nation- wide cohort.”
However, I do not believe this aim has been achieved. It is still unclear how many patients were currently diagnosed versus how many patients newly diagnosed with CP between 2016-2022.
It is also unclear how the prevalence of 0.9/1000 was measured/established.
Lines 181-191, It is unclear to me, but perhaps this is the main point of the manuscript, that the e-Nabiz program cannot answer the question regarding prevalence of CP and therefore a CP registry should be formed in Turkey. If these statements are accurate, then this paragraph should be rewritten.
Overall quality of English language is very good. There are some minor spelling, grammatic and typographical errors that should be addressed.
Author Response
We thank you for your thoughtful suggestions and insights. The manuscript has benefited from these insightful suggestions. The manuscript has been rechecked, and the necessary changes have been made following your recommendations. The responses to all comments have been prepared and are given in the Word file.
Q1: Lines 43-44 state: “Understanding the characteristics and prevalence of CP is crucial for its management and prevention”
Answer: The sentence has been revised in accordance with the recommendations provided.
‘Comprehending the attributes and prevelance of cerebral palsy (CP) holds significant importance in effectively managing the potential comorbidities that may arise as a result of this condition.’
Q2: Lines 52-53 state: “The body of literature addressing the comprehensive landscape of orthopaedic surgical procedures performed for children with CP is limited.”
Answer: The sentence has been revised in accordance with the recommendations provided.
‘Although studies involving nationwide cohorts related to cerebral palsy are, limited studies evaluating orthopaedic surgical interventions in these patients at the level of a nationwide cohort has been found in the literature.’
Q3: Lines 69-74: How is Turkish Ministry of Health’s database not part of a registry…possibly expound on this database and explain what it is and is not and why a separate CP registry is needed.
Answer: The related question was also raised by a different reviewer and the answer to this question is given below in the discussion section.
‘Even though GMFCS scores are documented within the health institution's data of the patient's follow-up in our country, it is regrettable that there is a lack of a distinct data tab within the e-Nabız system, which would enable us to record this information for the entire national cohort that we evaluate. Consequently, the GMFCS scores of the patients within our study cohort needed to be more attainable. Despite its significance in patient follow-up, the e-Nabız system is currently in its nascent stage with regards to the follow-up of individuals with cerebral palsy. Regrettably, the current documentation remains inadequate in capturing crucial data points such as the GMFCS scores of patients, the anatomical regions targeted by surgical interventions, the various subgroups of surgical interventions, and any complications that may arise from the use of the e-Nabız system.’
Q4: Lines: 85-86. Please expand on what criteria were used to decide, “lacking sufficient data”.
Answer: The sentence has been revised in accordance with the recommendations provided.
‘Patients who did not have sufficient data, such as limited data transfer to e-Nabız from the health institution they applied to, and those who did not allow open access to e-Nabız data, were excluded from the study.’
Q5: Line 87-91. Please clarify and re-write this paragraph. First sentence reports 184311 patients were diagnosed with CP.
Please clarify were these new diagnosis or already diagnosed.
The next sentence says between 2016-2022 53,027 children were born and 9658 had surgery.
This paragraph needs to be re-written to clarify how you identified: 1) how many children born between 2016 and 2022 were diagnosed with CP
2) at what age were they diagnosed CP
3) did all of the 9658 children that underwent surgery also previously have a diagnosis of CP?
Answer: According to your suggestions, the following sentences has been added to the method section.
‘During the period spanning from 2016 to 2022, a cumulative sum of 184311 patients diagnosed with CP was identified, and their respective cases were documented within the e-Nabız system. Among the patient population, 53027 individuals were identified who were born within the time frame of 2016 to 2022.’
‘As of 2016, the e-Nabız registration system data in our country is accessible with the highest degree of accuracy. Hence, we utilized the dataset of individuals born within the timeframe of 2016 to 2022 to conduct incidence and prevalence analyses, and to more precisely assess the demographic characteristics of patients who have been diagnosed with cerebral palsy. However, while giving information about surgical interventions, we evaluated the surgical procedures performed at the same time. To clarify, individuals who received a diagnosis of cerebral palsy prior to 2016 were eligible to participate in the surgical interventions conducted from 2016 to 2022.’
Q6: Lines 105-6: please clarify what ‘deformity correction’ surgical interventions are.
Answer: According to your suggestions, the following sentences has been added to the method section.
‘deformity correction ‘(distal femoral extension osteotomies, proximal femur derotation osteotomies, medial column corrections of the foot, etc.)’
Q7: Table 1:
What does the p value in the third column indicate?
How many of these children born were diagnosed with CP between 2016-2022?
Answer: All of the patients with a diagnosis of CP in Table 1 were born between the years 2016-2022. The p value in the 3rd column emphasizes that there is a statistically significant difference between the genders of the patients, the geographical regions where they were diagnosed, and the health institutions where they were diagnosed.
Q8: Figure 1: please explain what vertical axes represent. Is this a percentage?
Answer:Thank you for bringing this to our attention. After your warnings, we realized that we made a mistake in our incidence and prevalence calculations. We reviewed our data with professional support and made our calculations again.
We recalculated our annual incidences between the years 2016-2022 and included the incidences of every 10000 individuals under the age of 18 in the figure with the new Figure1.
It is still unclear how many persons were newly diagnosed with CP between 2016-2022.
Answer: Following sentences added in the materials and methods section.
‘During the period spanning from 2016 to 2022, a cumulative sum of 184311 patients diagnosed with CP was identified, and their respective cases were documented within the e-Nabız system. Among the patient population, 53027 individuals were identified who were born within the time frame of 2016 to 2022.’
If there were a lot of refugees included during this time, then it is still unclear how many persons were newly diagnosed with CP versus persons already diagnosed with CP and immigrated to Turkey. This information would greatly affect prevalence of CP and must be accounted for in some manner.
Answer: Following sentences added in the discussion section.
‘The influx of refugees to our nation experienced an escalation subsequent to 2012. From this year, the national health system has assigned a registration number to legally recognized asylum seekers, enabling them to access free healthcare services. Regrettably, the extent to which these children can benefit from health services remains uncertain due to various social challenges and impediments, including linguistic barriers. We believe that the aforementioned scenario does not impact the calculations of incidence and prevalence.’
Q9: Lines 135-143 and Table 2. Please clarify the 9658 patients, all were previously diagnosed with CP?
Answer: According to your suggestions, the following sentences has been added to the method section.
‘As of 2016, the e-Nabız registration system data in our country is accessible with the highest degree of accuracy. Hence, we utilized the dataset of individuals born within the timeframe of 2016 to 2022 to conduct incidence and prevalence analyses, and to more precisely assess the demographic characteristics of patients who have been diagnosed with cerebral palsy. However, while giving information about surgical interventions, we evaluated the surgical procedures performed at the same time. To clarify, individuals who received a diagnosis of cerebral palsy prior to 2016 were eligible to participate in the surgical interventions conducted from 2016 to 2022.’
Also, if 1.5 % of these patients are immigrants, then it seems to me these could be excluded if you are trying to say something about prevalence of CP.
Answer: We found that the rate of immigrant children born between 2016-2022 and diagnosed with cerebral palsy, which we included in the study to evaluate epidemiological characteristics, was 8.8%, while the rate of immigrants who underwent surgical intervention was 1.5%. These children are a part of our society, as they already carry a health registration number in our country, and they are trying to access free health care. Investigating why they have less access to surgical procedures is the subject of different professional studies and this is discussed in the discussion section.
Q10: Lines 154-5 state “In this study, we aimed to evaluate the prevalence and incidence of CP in a nation- wide cohort.”
However, I do not believe this aim has been achieved. It is still unclear how many patients were currently diagnosed versus how many patients newly diagnosed with CP between 2016-2022.
Answer: Following sentences added in the materials and methods section.
‘During the period spanning from 2016 to 2022, a cumulative sum of 184311 patients diagnosed with CP was identified, and their respective cases were documented within the e-Nabız system. Among the patient population, 53027 individuals were identified who were born within the time frame of 2016 to 2022.’
It is also unclear how the prevalence of 0.9/1000 was measured/established.
Answer: Thank you for bringing this to our attention. After your warnings, we realized that we made a mistake in our incidence and prevalence calculations. We reviewed our data with professional support and made our calculations again.
As of the year 2022, the number of individuals who have been diagnosed with CP and are under the age of 18 in our nation is 184311. The aggregate count of individuals within our populace who are below the age of 18 is 23823305. As per your information, the incidence rate among children under 18 is 7.74 per 1000. Thank you for your warnings and contributions.
Q11: Lines 181-191, It is unclear to me, but perhaps this is the main point of the manuscript, that the e-Nabiz program cannot answer the question regarding prevalence of CP and therefore a CP registry should be formed in Turkey. If these statements are accurate, then this paragraph should be rewritten.
Answer: In the discussion part, this issue was mentioned in line with your suggestions.
‘Despite its significance in patient follow-up, the e-Nabız system is currently in its nascent stage with regards to the follow-up of individuals with cerebral palsy. Regrettably, the current documentation remains inadequate in capturing crucial data points such as the GMFCS scores of patients, the anatomical regions targeted by surgical interventions, the various subgroups of surgical interventions, and any complications that may arise from the use of the e-Nabız system.’

Round 2
Reviewer 2 Report
Thank you for the additions and clarifications to the paper.
The title is now more accurate and the information on the different regions of Turkey add to clarity for the reader.
Unfortunately, there are still inconsistencies and inaccuracies in the reporting of your sample.
The paragraph added does explain that all surgical procedures conducted between 2016 and 2022 were included in this analysis, while the incidence and total population descriptions conducted only include children born between 2016 and 2022. However, this is still not clear and unambiguous as written. It is not accurate to say that individuals were eligible to participate in the surgical interventions. This sounds like they were participating in a study or clinical trial. This is a retrospective report of all individuals with CP who underwent surgery in the time period. Please rewrite that section.
In addition, the abstract states ‘53027 were born... and 9658 of them underwent surgery’ - this is incorrect. 9658 individuals under 18 years with CP in Turkey underwent surgery between 2016 and 2022.
Incidence and prevalence. The addition of the paragraph clarifying the difference between incidence and prevalence is helpful - and the details of the calculation of incidence and prevalence rates. The change from 0.9/1000 to 7.74/1000 makes much more sense.
The high prevalence rate in Turkey - in comparison to international estimates for both high, middle and lower resourced settings - requires more discussion. Although influenced by migration - since only 8.8% of the sample are migrants, this is not the whole answer.
Also - for clarification - when you say 8.8% of the sample are migrants - is this the 2016 to 2022 birth year sample? 1.5% of the surgical sample were migrants - but it is unknown how this relates to the numbers of migrants included in the total population of individuals with CP under 18 years.
I also wonder when you report incidence rates - for migrant children, does the year mean the year they were diagnosed, or their birth year, if they were born elsewhere? As you state, children with more severe CP are often diagnosed by 6 months of age, whereas GMFCS I and II may be older
When you state that the reduced number of hospital appointments during the COVID period is the main reason for reduced incidence - you are suggesting that children (probably with milder forms of CP) have not been seen for diagnosis yet? - and that if they were diagnosed later (at 2-3 years old for example) - their numbers would be added into the birth year data in any future analyses - is this correct?
GMFCS scores - thank-you for adding clarification as to why you were not able to extract that data from the eNabiz system. However, it would be more accurate to state that you were unable to include that data in this analysis for these reasons - rather than to say that data needs to be more attainable - and to acknowledge why GMFCS scores would have been important to include. It would also be helpful to indicate earlier (in methods) that GMFCS scores could not be extracted -rather than leaving the reader to wonder and only clarifying in the limitations or discussion sections.
Age bands - I understand the point you are making about GMFCS level and age of surgeries. However, you have 4 age bands - and have not clarified why you divided them as such with justification from the literature. Children up to 6 years are included in the more comprehensive analysis - and I understand you are splitting those into older and younger children. Is there a rationale related to timing of surgeries for dividing at 4-6 - or is it an arbitrary or convenient cutoff? This should be clarified. Same for the older age group not included in the main analysis - is there a rationale for under 10 years versus over?
Statistical analyses
Thank-you for adding the footnotes to table 1. Now I understand where those analyses were conducted. However, the larger question is why? With such large numbers, statistically significant differences are not unexpected. What is the importance of these? Unless you conducted post-hoc tests and use the differences to explain your results, then I don’t see the relevance.
There are more males than females - this is in line with international data as is somewhat mentioned in the discussion.
There are differences between geographical region and the type of setting where the diagnosis was made and the regions where surgeries are conducted.
Where are the Kruskal-Wallis test results reported?
I think you are pointing out that areas with higher incidences of CP should have higher incidences of surgeries - and this is not the case due to geographical and resource factors. This point should be made more clearly in results and then expanded in the discussion.
Overall there are only minor issues with English language. However, it is difficult to explain the differences in the two samples, and this requires more clarity.
Author Response
Dear reviewer, we express our gratitude for your assistance and recommendations in enhancing our research, rendering it more comprehensible, and augmenting its contribution to the existing body of literature. The responses to your inquiries have been appended.
Thank you for the additions and clarifications to the paper.
The title is now more accurate and the information on the different regions of Turkey add to clarity for the reader.
Unfortunately, there are still inconsistencies and inaccuracies in the reporting of your sample.
The paragraph added does explain that all surgical procedures conducted between 2016 and 2022 were included in this analysis, while the incidence and total population descriptions conducted only include children born between 2016 and 2022. However, this is still not clear and unambiguous as written. It is not accurate to say that individuals were eligible to participate in the surgical interventions. This sounds like they were participating in a study or clinical trial. This is a retrospective report of all individuals with CP who underwent surgery in the time period. Please rewrite that section.
In addition, the abstract states ‘53027 were born... and 9658 of them underwent surgery’ - this is incorrect. 9658 individuals under 18 years with CP in Turkey underwent surgery between 2016 and 2022.
Answer: To address the prevailing ambiguity, the paragraph has been revised in accordance with the provided recommendations.
‘As of 2022, the total number of children below the age of 18 who have been diagnosed with cerebral palsy in our nation amounts to 184,311. The prevalence calculation was conducted utilizing the provided dataset. In order to calculate the incidence of cerebral palsy, the researchers utilized the ratio between the number of newly diagnosed patients with cerebral palsy registered with e-Nabız from 2016 to 2022 and the corresponding population under the age of 18 for each respective year. The study, it was aimed to evaluate various demographic characteristics, including the geographical region where the diagnosis was made, by using the data set consisting of 53,027 people born in our country and diagnosed with cerebral palsy between the years 2016-2022. The main reason for this situation is that regular data transfer to e-Nabız, the patient registration system of our country, started in 2016. Furthermore, an analysis was conducted on the data of 9658 children who underwent surgical intervention between the years 2016 and 2022. It should be noted that the data from the registry system was easily accessible during this period.’
Incidence and prevalence. The addition of the paragraph clarifying the difference between incidence and prevalence is helpful - and the details of the calculation of incidence and prevalence rates. The change from 0.9/1000 to 7.74/1000 makes much more sense.
The high prevalence rate in Turkey - in comparison to international estimates for both high, middle and lower resourced settings - requires more discussion. Although influenced by migration - since only 8.8% of the sample are migrants, this is not the whole answer.
Answer: Following sentences were added in the discussion section for your suggestions:
‘Despite a general downward trend in the birth rate within our nation, 1,035,795 live births were recorded in 2022. This figure encompasses children of immigrants born within our country's borders. The high prevalence of cerebral palsy in regions with high birth rates, specifically eastern and southeastern Anatolia, can be attributed to various factors, including limited access to health services, home births, challenges in accessing paediatric intensive care units, and consanguineous marriages.’
Also - for clarification - when you say 8.8% of the sample are migrants - is this the 2016 to 2022 birth year sample? 1.5% of the surgical sample were migrants - but it is unknown how this relates to the numbers of migrants included in the total population of individuals with CP under 18 years.
Answer: Of the 53027 children whose demographic data we analyzed, 8.8% were immigrant children born between 2016 and 2022. 9658 Immigrant children constituted 1.5% of children under the age of 18 who underwent surgical intervention.
I also wonder when you report incidence rates - for migrant children, does the year mean the year they were diagnosed, or their birth year, if they were born elsewhere? As you state, children with more severe CP are often diagnosed by 6 months of age, whereas GMFCS I and II may be older
When you state that the reduced number of hospital appointments during the COVID period is the main reason for reduced incidence - you are suggesting that children (probably with milder forms of CP) have not been seen for diagnosis yet? - and that if they were diagnosed later (at 2-3 years old for example) - their numbers would be added into the birth year data in any future analyses - is this correct?
Answer: Following sentences added in the Results section
‘While calculating our incidence rates, we calculated by dividing the number of children registered to the e-Nabız system by the diagnosis of cerebral palsy in the year in which the incidence was calculated, to the population under the age of 18 in that year.’
This was independent of whether the child was an immigrant or not.
GMFCS scores - thank-you for adding clarification as to why you were not able to extract that data from the eNabiz system. However, it would be more accurate to state that you were unable to include that data in this analysis for these reasons - rather than to say that data needs to be more attainable - and to acknowledge why GMFCS scores would have been important to include. It would also be helpful to indicate earlier (in methods) that GMFCS scores could not be extracted -rather than leaving the reader to wonder and only clarifying in the limitations or discussion sections.
Answer: The following sentences are included in the methods section.
‘The primary rationale behind categorizing the patients based on their age pertains to the unavailability of the GMFCS scores of the patients. The grouping of patients based on age can be informative in assessing surgical interventions, given that those with a higher GMFCS score tend to require such procedures at an earlier stage.’
Age bands - I understand the point you are making about GMFCS level and age of surgeries. However, you have 4 age bands - and have not clarified why you divided them as such with justification from the literature. Children up to 6 years are included in the more comprehensive analysis - and I understand you are splitting those into older and younger children. Is there a rationale related to timing of surgeries for dividing at 4-6 - or is it an arbitrary or convenient cutoff? This should be clarified. Same for the older age group not included in the main analysis - is there a rationale for under 10 years versus over?
Answer: As previously indicated, the primary rationale behind categorizing age groups in the prescribed manner is the inability to obtain the Gross Motor Function Classification System (GMFCS) scores. The following sentences were added in the methods section.
‘During establishing these groups, we derived valuable insights from the research conducted by Telleus et al. in their publication of 2022. Nevertheless, due to the utilization of Swedish registry data and the incorporation of patients' GMFCS scores, the direct applicability of this study was limited. When the conclusions derived from this study are integrated with our routine pediatric orthopedic experiences, it becomes evident that a surge in surgical interventions is observed after the initial three years of age, irrespective of GMFCS scores. Furthermore, there is a notable rise in soft tissue operations and BTX injections between the ages of 4-6. Additionally, surgical interventions aimed at averting permanent deformities are more prevalent during the ages of 7-10. Given that our focus primarily revolves around deformity correction and joint-sparing surgeries for individuals aged ten years and above, we have proceeded with the categorization process.’
Statistical analyses
Thank-you for adding the footnotes to table 1. Now I understand where those analyses were conducted. However, the larger question is why? With such large numbers, statistically significant differences are not unexpected. What is the importance of these? Unless you conducted post-hoc tests and use the differences to explain your results, then I don’t see the relevance.
Answer: Although we can observe the difference when we look at the numbers observationally, we wanted to share the statistical data. We appreciate your understanding.
There are more males than females - this is in line with international data as is somewhat mentioned in the discussion.
There are differences between geographical region and the type of setting where the diagnosis was made and the regions where surgeries are conducted.
Where are the Kruskal-Wallis test results reported?
Answer: We used the Kruskal Wallis test in the part where we examined the relationship between age groups and surgical procedures, but we did not share it as data to avoid confusion. Therefore, we remove this part from the methods in line with your suggestions.
I think you are pointing out that areas with higher incidences of CP should have higher incidences of surgeries - and this is not the case due to geographical and resource factors. This point should be made more clearly in results and then expanded in the discussion.
Answer: Following sentences were added in the discussion section for your suggestions.
‘The findings of our study indicate that the diagnosis of cerebral palsy is predominantly concentrated in the southeastern region. This region is situated in predominantly rural areas and is distinguished as the geographical region exhibiting the highest birth rate within our nation. However, considering the volume of surgical procedures conducted, it becomes evident that urban areas take precedence.’

Reviewer 3 Report
This is the second round of review for a paper about the prevalence of cerebral palsy (CP) in Turkish children and how many had different types of surgery or Botulinum Toxin injections. In my previous review, I raised a number of issues which I thought could be addressed. Despite the authors’ changes, most of my concerns still remain. Moreover, the changes raise some new concerns. The main ones are as follows:
1. As this paper is a report of the prevalence of CP in one country, it is relevant to understand factors that might influence the prevalence in that country. Therefore, it would be useful in the Introduction to have background about diagnosis of CP in Türkiye (e.g., age of diagnosis, processes that might influence who is diagnosed). The authors have responded by adding some general information about CP diagnosis, but not specific to Türkiye.
2. The following two sentences have been added to this version of the paper: “During the period spanning from 2016 to 2022, a cumulative sum of 184311 patients diagnosed with CP was identified, and their respective cases were documented within the e-Nabız system. Among the patient population, 53027 individuals were identified who were born within the time frame of 2016 to 2022.” Do you mean that there were 184311 children in Türkiye aged between 0 and 18 years with a diagnosis of CP? And that 53027 were born in the years 2016-2022? At what point in the period 2016-2022 were there 184311 children? In other words, the number was presumably changing all the time as new children were diagnosed and older children turned 18. So at what point did you calculate the 184311 children?
3. Since the last version of this paper, the authors have identified an error in their prevalence figure, which is not 0.9 but 7.74. Given this error, could the authors please give the raw figures (e.g., 184311/23823305) as well as the prevalence and incidence figures (e.g., 7.74 per 1000), so that the readers can be sure of the data? To do this, year by year, perhaps Figure 1 could be replaced by a table. (And please correct the spelling of “incidence” in that figure when converted to a table.)
4. Why is the prevalence figure of 7.74/1000 given to 2 decimal places and out of 1000, while the yearly incidence figures are given with 0 decimal places and out of 10,000?
5. Please change “23.823.305 children” to “23 823 305 children” or “23,823,305 children”.
6. The prevalence figure is high (7.74/1000), whereas the yearly incidence figures are low (ranging from 0.4/1000 to 1/1000). How do the authors explain large difference between them? The apparent discrepancy makes me ask: are these figures now correct?
7. The data on the surgical procedures are quite limited. There is no breakdown of types of surgeries (e.g., by area of the body) nor any data about the number of botulinum injections each child had.
8. In the Discussion section, there is no comparison of the prevalence and incidence of CP found in the present study with those of other studies. Nor is there any consideration as to whether it is the same as other countries or why it might be higher or lower.
English is good.
Author Response
Dear reviewer, we express our gratitude for your assistance and recommendations in enhancing our research, rendering it more comprehensible, and augmenting its contribution to the existing body of literature. The responses to your inquiries have been appended.
This is the second round of review for a paper about the prevalence of cerebral palsy (CP) in Turkish children and how many had different types of surgery or Botulinum Toxin injections. In my previous review, I raised a number of issues which I thought could be addressed. Despite the authors’ changes, most of my concerns still remain. Moreover, the changes raise some new concerns. The main ones are as follows:
- As this paper is a report of the prevalence of CP in one country, it is relevant to understand factors that might influence the prevalence in that country. Therefore, it would be useful in the Introduction to have background about diagnosis of CP in Türkiye (e.g., age of diagnosis, processes that might influence who is diagnosed). The authors have responded by adding some general information about CP diagnosis, but not specific to Türkiye.
Answer: Following sentences were added in the introduction section.
In our country, the initial stage of diagnosing cerebral palsy (CP) involves the contribution of neonatal physicians who assess the risk factors during the perinatal period. Following this timeframe, family physicians and pediatricians facilitate the referral of patients to pediatric neurologists for the purpose of obtaining additional diagnostic assessments to confirm the presence of the disease. Following the establishment of a conclusive diagnosis, the patient's treatment plan typically involves medical professionals specializing in physical therapy, orthopedics, and neurosurgery.
- The following two sentences have been added to this version of the paper: “During the period spanning from 2016 to 2022, a cumulative sum of 184311 patients diagnosed with CP was identified, and their respective cases were documented within the e-Nabız system. Among the patient population, 53027 individuals were identified who were born within the time frame of 2016 to 2022.” Do you mean that there were 184311 children in Türkiye aged between 0 and 18 years with a diagnosis of CP? And that 53027 were born in the years 2016-2022? At what point in the period 2016-2022 were there 184311 children? In other words, the number was presumably changing all the time as new children were diagnosed and older children turned 18. So at what point did you calculate the 184311 children?
Answer: To address the prevailing ambiguity, the paragraph has been revised in accordance with the provided recommendations.
‘As of 2022, the total number of children below the age of 18 who have been diagnosed with cerebral palsy in our nation amounts to 184,311. The prevalence calculation was conducted utilizing the provided dataset. In order to calculate the incidence of cerebral palsy, the researchers utilized the ratio between the number of newly diagnosed patients with cerebral palsy registered with e-Nabız from 2016 to 2022 and the corresponding population under the age of 18 for each respective year. The study, it was aimed to evaluate various demographic characteristics, including the geographical region where the diagnosis was made, by using the data set consisting of 53,027 people born in our country and diagnosed with cerebral palsy between the years 2016-2022. The main reason for this situation is that regular data transfer to e-Nabız, the patient registration system of our country, started in 2016. Furthermore, an analysis was conducted on the data of 9658 children who underwent surgical intervention between the years 2016 and 2022. It should be noted that the data from the registry system was easily accessible during this period.’
3.Since the last version of this paper, the authors have identified an error in their prevalence figure, which is not 0.9 but 7.74. Given this error, could the authors please give the raw figures (e.g., 184311/23823305) as well as the prevalence and incidence figures (e.g., 7.74 per 1000), so that the readers can be sure of the data? To do this, year by year, perhaps Figure 1 could be replaced by a table. (And please correct the spelling of “incidence” in that figure when converted to a table.)s
Answer: In line with your suggestions, the figure was removed and a table containing the incidences was added.
|
Years |
Newly Diagnosed CP |
Population under 18 |
Incidence per 1000 |
|
2016 |
25634 |
24 223 372 |
1.05 |
|
2017 |
24870 |
24 224 739 |
1.03 |
|
2018 |
20928 |
24 258 266 |
0.86 |
|
2019 |
17762 |
24 196 448 |
0.73 |
|
2020 |
10800 |
23 979 485 |
0.45 |
|
2021 |
11807 |
23 960 578 |
0.49 |
|
2022 |
11268 |
23 823 305 |
0.47 |
- Why is the prevalence figure of 7.74/1000 given to 2 decimal places and out of 1000, while the yearly incidence figures are given with 0 decimal places and out of 10,000?
Answer: Your suggested changes have been made.
‘ In the annual incidence calculations between 2016 and 2022, the annual incidence was 1.05/1000 in 2016, 1.03/1000 in 2017, 0.86/1000 in 2018, 0.73/1000 in 2019, 0.45/1000 in 2020, 0.49/1000 in 2021 and 0.47/1000 in 2022.’
- Please change “23.823.305 children” to “23 823 305 children” or “23,823,305 children”.
Answer: Your suggested changes have been made.
- The prevalence figure is high (7.74/1000), whereas the yearly incidence figures are low (ranging from 0.4/1000 to 1/1000). How do the authors explain large difference between them? The apparent discrepancy makes me ask: are these figures now correct?
Answer: The data we made the calculations and the results are clearly shared. We appreciate your understanding.
- The data on the surgical procedures are quite limited. There is no breakdown of types of surgeries (e.g., by area of the body) nor any data about the number of botulinum injections each child had.
Answer: Unfortunately, we cannot access this data clearly via our data e-pulse. We appreciate your understanding.
- In the Discussion section, there is no comparison of the prevalence and incidence of CP found in the present study with those of other studies. Nor is there any consideration as to whether it is the same as other countries or why it might be higher or lower.
Answer: The following sentences are included in the Discussion section.
CP is a disease that should be treated by teams of many healthcare professionals, including orthopaedic surgeons, as it requires a multidisciplinary approach, and it impairs quality of life due to symptoms such as joint deformities and muscle contractures. CP is the most common cause of motor disability in childhood, with an incidence of approximately 1.5-3.0/1000 live births [3,6,13]. Like most neurological disorders, it occurs appreciably more frequently among male patients than females[3]. In our study, we found that the prevalence of CP was 7.74/1000 and it was more common among male patients. Despite a general downward trend in the birth rate within our nation, 1 035 795 live births were recorded in 2022. This figure encompasses children of immigrants born within our country's borders. The high prevalence of cerebral palsy in regions with high birth rates, specifically eastern and southeastern Anatolia, can be attributed to various factors, including limited access to health services, home births, challenges in accessing pediatric intensive care units, and consanguineous marriages. The incidence of a disease is a metric that quantifies the frequency of its manifestation, while prevalence is a metric that quantifies its overall presence. The incidence metric identifies new cases, whereas the prevalence metric provides information on new and pre-existing cases. Since the inception of the e-Nabız patient registration system in 2016, the incidence data has been consistently recorded with high accuracy. Notably, the incidence rate was observed to be high during the years 2016 and 2017. However, in the years 2020 and 2021, the incidence rate has declined, which can be attributed to the limited number of hospital admissions resulting from the COVID-19 pandemic. This is thought to be the primary factor contributing to the reduction in occurrence.

Reviewer 4 Report
The authors have addressed all the questions the reviewer were concerning.
Author Response
Dear reviewer, we express our gratitude for your assistance and recommendations in enhancing our research, rendering it more comprehensible, and augmenting its contribution to the existing body of literature. The responses to your inquiries have been appended.
Sincerely.
Reviewer 5 Report
I would like to thank the authors for addressing each of my concerns. I know this is a very time consuming task. However, I believe the changes made have improved the manuscript and more importantly provides clear information for those of us not familiar with Turkey's health care system.
Author Response

(The authors gave the same response as above.)
